# Cognitive Effects of Simulated Galactic Cosmic Radiation Are Mediated by ApoE Status, Sex, and Environment in APP Knock-In Mice

**DOI:** 10.3390/ijms25179379

**Published:** 2024-08-29

**Authors:** Laura Wieg, Jason C. Ciola, Caroline C. Wasén, Fidelia Gaba, Brianna R. Colletti, Maren K. Schroeder, Robert G. Hinshaw, Millicent N. Ekwudo, David M. Holtzman, Takashi Saito, Hiroki Sasaguri, Takaomi C. Saido, Laura M. Cox, Cynthia A. Lemere

**Affiliations:** 1Department of Neurology, Ann Romney Center for Neurologic Diseases, Brigham and Women’s Hospital, Boston, MA 02115, USA; l.wieg@umcutrecht.nl (L.W.); jasonciola@gmail.com (J.C.C.); caroline.wasen@gu.se (C.C.W.); fig5@pitt.edu (F.G.); bcolletti21@gmail.com (B.R.C.); schroemk@odu.edu (M.K.S.); rhinshaw@alum.mit.edu (R.G.H.); m.ekwudo@florey.edu.au (M.N.E.); lcox@bwh.harvard.edu (L.M.C.); 2Harvard-MIT Division of Health Sciences and Technology, Massachusetts Institute of Technology, Cambridge, MA 02139, USA; 3Department of Neurology, Hope Center for Neurological Disorders, Knight Alzheimer’s Disease Research Center, Washington University School of Medicine, St. Louis, MO 63110, USA; holtzman@wustl.edu; 4Department of Neurocognitive Science, Institute of Brain Science, Nagoya City University Graduate School of Medical Science, Nagoya 467-8601, Aichi, Japan; saito-t@med.nagoya-cu.ac.jp; 5Laboratory for Proteolytic Neuroscience, RIKEN Center for Brain Science, Wako City 351-0198, Saitama, Japan; hiroki.sasaguri@riken.jp (H.S.); takaomi.saido@riken.jp (T.C.S.); 6Department of Neurology, Harvard Medical School, Boston, MA 02115, USA

**Keywords:** galactic cosmic radiation, gamma radiation, Alzheimer’s disease, sex differences, apolipoprotein E, gut microbiome, APP mouse models

## Abstract

Cosmic radiation experienced during space travel may increase the risk of cognitive impairment. While simulated galactic cosmic radiation (GCRsim) has led to memory deficits in wildtype (WT) mice, it has not been investigated whether GCRsim in combination with genetic risk factors for Alzheimer’s disease (AD) worsens memory further in aging mice. Here, we investigated the central nervous system (CNS) effects of 0 Gy (sham) or 0.75 Gy five-ion GCRsim or 2 Gy gamma radiation (IRR) in 14-month-old female and male APP^NL-F/NL-F^ knock-in (KI) mice bearing humanized ApoE3 or ApoE4 (APP;E3F and APP;E4F). As travel to a specialized facility was required for irradiation, both traveled sham-irradiated C57BL/6J WT and KI mice and non-traveled (NT) KI mice acted as controls for potential effects of travel. Mice underwent four behavioral tests at 20 months of age and were euthanized for pathological and biochemical analyses 1 month later. Fecal samples were collected pre- and post-irradiation at four different time points. GCRsim seemed to impair memory in male APP;E3F mice compared to their sham counterparts. Travel tended to improve cognition in male APP;E3F mice and lowered total Aβ in female and male APP;E3F mice compared to their non-traveled counterparts. Sham-irradiated male APP;E4F mice accumulated more fibrillar amyloid than their APP;E3F counterparts. Radiation exposure had only modest effects on behavior and brain changes, but travel-, sex-, and genotype-specific effects were seen. Irradiated mice had immediate and long-term differences in their gut bacterial composition that correlated to Alzheimer’s disease phenotypes.

## 1. Introduction

There is an increasing demand for deep space exploration in the coming decades, with the National Aeronautics and Space Administration (NASA) planning for lunar missions this decade and manned Mars voyages as early as the 2030s and with companies offering private individuals increasingly numerous opportunities for space tourism. While in deep space, astronauts will be exposed to low but continuous levels of galactic cosmic radiation (GCR), which consists of alpha particles, protons, and high-energy, high-charged (HZE) particles. GCR tends to be much more detrimental to living organisms than electromagnetic radiation typically found on Earth and in low orbit, e.g., gamma and X-rays, which are only sparsely ionizing. Some components of GCR (like protons and alpha particles) can be partially shielded, but HZE particles can traverse the spacecraft shielding or create showers of secondary radiation when interacting with shielding or human tissues [1]. It is expected that every nucleus of a human body in space encounters an HZE particle about once a month [2,3]. While natural sources of radiation lead to a worldwide average effective dose per capita on Earth of 2.4 mSv/year [4], NASA estimated that astronauts being on a Mars mission for 2.5 years would experience 1 Sievert or 0.28 Gy at a rate of 400 mSv/year [5,6]. Additionally, commercial airline flight crews, which represent a cohort many orders of magnitude larger, can experience annual effective total radiation doses up to 3.75 times higher than those experienced at sea level, with a much larger percentage coming from GCR due to the significant reduction in atmospheric shielding at common cruising altitudes [7]. Late risks on the central nervous system (CNS) cannot be discounted, especially when radiation effects are combined with other risk factors.

More recently, studies using a multi-ion irradiation approach, called simulated GCR (GCRsim), instead of single-ion radiation (reviewed in [8]) were conducted, providing a radiation spectrum that more closely resembles what astronauts will be exposed to while travelling through deep space. It has been shown that 5-beam (proton, ^16^O, ^4^He, ^28^Si, proton) and 6-beam (proton, ^28^Si, ^4^He, ^16^O, ^56^Fe, proton) GCRsim lowers the rate of action potentials and impairs hippocampal long-term potential (LTP) in wildtype (WT) mice, disturbing hippocampus-dependent memory [9,10]. Exposure to 0.5 Gy three-ion GCRsim leads to hippocampal synaptic loss in male WT mice, followed by less social interaction and recognition [11], while 0.5 Gy five-ion GCRsim causes spatial learning deficits [12]. Only recently, a study reported the effects of a new 33-beam GCRsim approach, delivered either chronically (0.5 Gy) or acutely (0.4 Gy), on 6-month-old male and female WT mice [13]. Persistent impairments in hippocampus-dependent memory were found in female mice, while recognition memory was impaired in males. In addition, synaptic plasticity in the hippocampal CA1 region was disrupted [13]. Studies on the effects of 0.1 Gy six-ion GCRsim on behavior in 7-month-old female and male Wistar rats were published recently [14,15,16]. In male rats, no significant deficits in sensorimotor function and motivation, temporal, and/or attentional processes were found 72 h after exposure to 0.1 Gy GCRsim compared to sham controls [16]. In contrast, GCRsim exposure induced a delay in problem solving in males [15]. Female rats exposed to 0.1 Gy six-ion GCRsim at 7 months old displayed impairments in problem solving when confronted with a high cognitive load [14].

Even for gamma rays, effects on the neural tissue of WT mice have been observed. In female mice, 0.04 Gy distributed over 21 days, led to long-term transcriptional alterations associated with increased neurogenesis and neuropeptide production [17]. Higher doses of gamma radiation have been shown to negatively affect neurogenesis in the dentate gyrus (1, 3, or 5 Gy) [18,19], cause higher reactive oxygen species (ROS) levels (2 Gy) [20], and decrease spatial and recognition memory (1 or 2.5 Gy) [21].

To our knowledge, no studies on the effects of multi-ion GCRsim on behavior and brain pathology in Alzheimer’s disease (AD)-like mouse models have been published to date. However, detrimental effects of irradiation with ^56^Fe, being the single largest particle in terms of biologically effective radiation received by astronauts in space [22], have been shown in APP23 and APP/PS1 mouse models for AD in the form of accelerated pathology [23,24,25,26]. APPswe/PSEN1dE9 (APP/PS1) mice exposed to 0.1 or 1 Gy ^56^Fe radiation at 9.5 (male) or 7 (female) months of age displayed impaired fear and recognition memory 6 months post-irradiation and elevated Aβ plaque levels in male mice [24]. Using the same mouse model in addition to WT mice, we have previously shown that irradiation with 0.1 or 0.5 Gy ^56^Fe ions at 4 months of age leads to sex-, genotype-, and dose-dependent changes in behavior 1.5 months later [25]. Motor coordination was improved only in female transgenic mice after irradiation with 0.5 Gy ^56^Fe, while male transgenic mice showed a significant decrease in contextual fear conditioning (CFC) when irradiated with 0.5 Gy in comparison to 0.1 Gy [25]. Radiation reduced fibrillar amyloid in the hippocampus (HC) of female transgenic mice [25]. Mice that underwent behavioral testing at 11 months instead of 5 months of age showed an effect of genotype on contextual fear learning and memory, representing long-term effects of ^56^Fe radiation on the CNS [26]. In male 3xTg mice, a decreased freezing response in CFC and a trend towards lower Aβ plaque levels was observed at 16 months of age compared to females and following irradiation with 0, 0.1, or 1 Gy ^56^Fe at 9 months of age [27].

With regard to astronauts exploring deep space, genetic risk factors for AD may interact with pathological effects of sex and radiation. Apolipoprotein E (ApoE) plays a major role. Of the three isoforms, the ApoE-ε4 (ApoE4) allele represents the strongest known genetic risk factor for late-onset AD (LOAD) [28] and can be found in approximately 40–60% of AD patients [29,30,31]. While carriers of one ApoE-ε4 allele have a 3-fold increased risk of developing AD, two alleles increase the risk by 12-fold compared to homozygous ApoE-ε3 (ApoE3) [32]. However, the association between ApoE-ε4 and AD risk differs between ethnicities [29,33,34,35,36,37], with a weaker association being reported for Blacks and Hispanics compared to Whites [33,35] and a higher risk for East Asians compared to Europeans [37]. ApoE protects the brain against iron radiation injuries, as shown in irradiated ApoE-deficient mice [11]. A connection between genotype, radiation, and oxidative stress is hypothesized [38,39].

The risk of developing AD is not only influenced by genetic factors but also by the gut microbiome, which again can be modulated by GCR, as described elsewhere [40]. Bidirectional communication between the gut and brain follows the microbiota–gut–brain axis, allowing the gut microbiome to modulate neuroinflammation, either directly (e.g., via the vagus nerve) or indirectly. Gut dysbiosis, meaning an altered composition of microbiota, has been shown for advanced disease stages in transgenic mouse models for AD [41,42,43,44,45,46]. In AD patients, an increase in pro-inflammatory taxa is typical, while anti-inflammatory taxa are decreased, with regional differences being detected in studies from different countries [47,48,49,50,51]. An association between dysbiosis, changed levels of circulatory biomarkers of inflammation, and altered levels of cerebrospinal fluid (CSF) Aβ and phospho-tau was reported previously [52]. Furthermore, it has been shown that the gut microbiome of ApoE4 carriers contains a lower number of bacteria that are protective in terms of gut barrier integrity [53].

In the present study, we irradiated 14-month-old male and female APP^NL-F/NL-F^ knock-in (KI) mice bearing humanized ApoE3 or ApoE4 (APP;E3F and APP;E4F) with 0 (sham) or 0.75 Gy of a simplified five-ion mixed-field beam (proton, ^28^Si, ^4^He, ^16^O, ^56^Fe) or 2 Gy of gamma rays. As travel to a specialized facility was required for irradiation, age-matched non-traveled (NT) APP;E3F and APP;E4F mice acted as controls for potential effects of travel and were compared to traveled sham-irradiated KI mice. In addition, sham-irradiated KI mice were compared to age-matched sham-irradiated C57BL/6J WT mice to detect genotype effects. Mice underwent behavioral testing at 20 months of age, and analyses of brain tissue and blood plasma were conducted at 21 months of age to examine late sex-/dose-/genotype-dependent radiation effects of low-dose GCRsim and gamma ray irradiation on the CNS. Fecal samples were collected pre- and post-irradiation to determine space radiation effects on the gut microbiome. In this study, we hypothesize that low-dose GCRsim and gamma rays will have detrimental effects on behavior and AD pathology in aged female and male APP;E3F, APP;E4F, and WT mice based on previous studies using single ions.

## 2. Results

### 2.1. Travel Resulted in Positive Effects on APP;E3F and APP;E4F Mouse Behavior, While Radiation and Sex Had Mixed Effects, and Female Mice Experienced Genotype Effects

This study utilized the open field (OF), rotarod, novel object recognition (NOR), and spatial novelty Y-maze (SNYM) tests to measure irradiation-, travel-, sex-, and genotype-induced behavioral changes. A total of 0.75 Gy GCRsim irradiation produced varying effects on different tests. Gamma-irradiated mice displayed no significant differences or trends compared to sham-irradiated mice. Surprisingly, sham-irradiated mice who traveled universally performed better or had no statistically significant difference in performance when compared to their non-traveled counterparts’ behavior tests results. No clear effects of sex and genotype were found.

There were no effects of radiation, travel, sex, or genotype on the total ambulatory distance the mice traveled nor the percentage time spent in the center of the OF for the 30 min test duration, indicating similar capacities for locomotion and anxiety-like behavior (Figure 1A–F). To exclude the effects of adaptation to the environment, only the first 5 min were analyzed. No significant irradiation effects were observed during the first 5 min on the OF test (Figure 1G). However, it was observed that there was significantly less anxiety-like behavior in traveled female APP;E4F mice who were sham-irradiated compared to non-traveled female controls (Figure 1H, *p* = 0.0268). A trend was observed between female and male APP;E4F mice, wherein females displayed less anxiety-like behavior than males (Figure 1I, *p* = 0.0986). One genotype effect was observed, where female sham-irradiated APP;E4F mice had significantly less anxiety-like behavior than female sham-irradiated APP;E3F mice (Figure 1I, *p* = 0.0411).

A radiation effect was found between sham- and 0.75 Gy GCRsim-irradiated female APP;E4F mice during the rotarod test, with the irradiated group showing better motor coordination by, on average, remaining on the rotarod for significantly more time than the sham controls (Figure 1J, *p* = 0.0031). No significant effect was found between any sham-irradiated group and their 2 Gy gamma-irradiated counterparts (Figure 1J). No significance or trend was found between any non-traveled and traveled sham-irradiated groups (Figure 1K). A trend was observed between sham-irradiated male and female APP;E3F mice, with the female mice having a higher average latency to fall (Figure 1L, *p* = 0.0870). All sham-irradiated female groups had a higher average time on the rotarod than any single sham-irradiated male group, while no genotype effects were seen for the average time on the rotarod (Figure 1L). No significant irradiation effect was observed between irradiated groups regarding the improvement between rotarod trials (Figure 1M). Between rotarod trials, sham-irradiated female APP;E3F mice increased their performance significantly more than their non-traveled counterparts, indicating an increased motor learning ability (Figure 1N, *p* = 0.0319). The 0.75 Gy GCRsim-irradiated female APP;E4F mice showed no improvement in performance between trials; however, this may have been due to their high average performance during the first trial (Figure 1M). A trend showing superior motor learning capabilities in sham-irradiated WT male mice compared to sham-irradiated female mice was observed (Figure 1O, *p* = 0.0988). No significant genotype effects or trends were found for the change in performance between rotarod trials (Figure 1O).

Our hypothesis that irradiation would affect long-term memory on the NOR was not confirmed. There were no effects of travel, irradiation, sex, or genotype on the percentage time spent exploring the novel object after a 24 h inter-trial interval (Figure 2A–C).

The SNYM test was used to investigate both locomotion (Figure 2D–F) as well as short-term spatial memory after an inter-trial interval of 2 min (Figure 2G–I).

For locomotive behavior, a significant difference was found between the sham-irradiated male APP;E3F mice and 0.75 Gy GCRsim-irradiated male APP;E3F mice, with the irradiated mice traveling more in the novel arm of the SNYM (Figure 2D, *p* = 0.0394). No effects on this metric were observed between any sham-irradiated groups and their 2 Gy gamma-irradiated counterparts (Figure 2D). Sham-irradiated female APP;E3F mice traveled significantly more in the novel arm than their non-traveled counterparts, indicating increased locomotion (Figure 2E, *p* = 0.0307). Two sex effect trends were discovered; for sham-irradiated APP;E3F mice, females trended towards increased locomotion compared to male counterparts, while the opposite was observed for sham-irradiated APP;E4F mice (Figure 2F, *p* = 0.0859 and 0.0983, respectively). Female sham-irradiated WT and APP;E3F mice traveled significantly more than female APP;E4F counterparts (Figure 2F, *p* = 0.0314 and *p* = 0.0082, respectively), displaying a genotype effect.

Evidence supporting our hypothesis for radiation-induced effects on cognition was observed, as a trend towards worse short-term memory in 0.75 Gy GCRsim-irradiated male APP;E3F mice compared to their sham-irradiated counterparts was observed (Figure 2G, *p* = 0.0765). Another trend was indicated between sham-irradiated male APP;E3F mice and their non-traveled counterparts, with the traveled mice spending more time exploring the non-familiar arm of the maze (Figure 2H, *p* = 0.0579). No sex or genotype effects were seen for this metric (Figure 2I).

### 2.2. Travel Resulted in Mostly Positive Effects on APP;E3F and APP:E4F Mouse Pathology, While Radiation and Sex Had No Significant Effects, and Mice of Both Sexes Experienced Genotype Effects

There were no significant effects of sex or irradiation on brain pathology. Pathological non-traveled vs. travel significances and trends largely followed the same inclination as behavior results, with two trends and a strong significance indicating that non-traveled mice had more microhemorrhages and Aβ pathology than their sham-irradiated traveled counterparts. However, one instance arose in which the traveled mice showed a significantly worse pathology. However, ApoE KI mice typically had more microhemorrhages and, unsurprisingly, more Aβ pathology than their WT counterparts.

No irradiation effects on microhemorrhages were found across whole-hemibrain sections (Figure 3A; Appendix A). Sham-irradiated male APP;E4F mice showed a trend towards fewer microhemorrhages than their non-traveled counterparts (Figure 3B, *p* = 0.0693). Unlike all previous non-traveled vs. travel results, sham-irradiated female APP;E3F mice had significantly more microhemorrhages than their non-traveled counterparts (Figure 3B, *p* = 0.0318). No effects of sex on microhemorrhages were found (Figure 3C). Sham-irradiated female APP;E3F mice had significantly more (*p* = 0.0125 and *p* = 0.0228, respectively) microhemorrhages than both their WT and APP;E4F counterparts, and sham-irradiated male APP;E4F mice had a trend for more microhemorrhages than their WT counterparts (Figure 3C, *p* = 0.0727).

Total Aβ plaque load in the hippocampus (HC) was quantified as the percentage region of interest (ROI) after immunolabeling brain slices with the anti-Aβ antibody S97 (Figure 3D–F; Appendix A). No significant irradiation effects were found (Figure 3D). As seen with almost all other metrics, non-traveled female APP;E3F mice had significantly worse hippocampal Aβ pathology than their traveled counterparts (Figure 3E, *p* = 0.0091). Non-traveled male APP;E3F mice similarly showed a trend for more Aβ pathology in the HC than those that traveled (Figure 3E, *p* = 0.0750). Female mice showed very significant differences for hippocampal plaque load between sham-irradiated WT mice and their APP;E3F and APP;E4F counterparts (*p* = 0.0076 and *p* = 0.0212, respectively), with the plaque load being significantly lower in WT mice, as expected (Figure 3F). Similarly, male mice displayed significantly lower amounts of total hippocampal Aβ load in sham-irradiated WT mice compared to their APP;E3F and APP;E4F counterparts (*p* = 0.0199 and *p* = 0.0168, respectively; Figure 3F).

In addition, the amount of fibrillar amyloid was measured in the HC as the percentage ROI after staining brain slices with the β-sheet-structured dye Thioflavin S (ThiosS) (Figure 3G–I; Appendix A). Compared to total Aβ (Figure 3D–F), the percentage ROI of fibrillar amyloid was much lower (Figure 3G–I). No significant effects of irradiation (Figure 3G), travel (Figure 3H), or sex (Figure 3I) on fibrillar amyloid load were found. A significant (*p* = 0.0176) and very significant (*p* = 0.0059) difference was found between female WT mice and their APP;E3F and APP;E4F counterparts, respectively, where the KIs displayed more fibrillar amyloid pathology (Figure 3I). Similarly, male mice showed a trend (*p* = 0.0668) and a significance (*p* = 0.0127) between WT and their APP;E3F and APP;E4F counterparts, with a higher hippocampal fibrillar amyloid load in the KI mice (Figure 3I). Male APP;E4F mice also had significantly more fibrillar amyloid pathology than their APP;E3F counterparts (Figure 3I, *p* = 0.0311).

### 2.3. GCRsim Radiation Decreased Total and HDL Cholesterol Levels in the Plasma, While Gamma Radiation Increased Triglyceride Levels

Application of Fujifilm assay kits on plasma of GCRsim- or gamma-irradiated APP;E3F and APP;E4F mice revealed effects on total and HDL cholesterol and triglyceride concentrations 7 months post-irradiation (Figure 4A–I).

Total cholesterol level was affected by 0.75 Gy GCRsim radiation, which was seen as a trend towards a lower concentration in male APP;E3F mice compared to sham controls (Figure 4A, *p* = 0.072) and a trend for an increase in concentration in female APP;E3F 2 Gy gamma-irradiated mice compared to sham controls (Figure 4A, *p* = 0.0795). No effects of travel on total cholesterol level were found (Figure 4B), but genotype had an influence. Male APP;E3F mice showed higher concentrations of total cholesterol compared to APP;E4F (*p* = 0.0405) and WT (*p* = 0.076) mice and compared to their female counterparts (Figure 4C, *p* = 0.018). High-density lipoprotein (HDL) cholesterol concentration was significantly decreased in male APP;E4F mice that were irradiated with 0.75 Gy GCRsim compared to sham controls (Figure 4D, *p* = 0.0461). No travel effects were seen (Figure 4E). HDL cholesterol concentration also showed a trend towards a higher concentration in male WT mice compared to their female counterparts (Figure 4F, *p* = 0.0616), but no genotype effects were detected (Figure 4F). Triglyceride concentration was significantly increased by 2 Gy gamma radiation in female APP;E3F mice (*p* = 0.0128) and showed a trend towards an increase in female APP;E4F mice (*p* = 0.0967) compared to sham-irradiated counterparts (Figure 4G). Interestingly, this was not observed in male APP;E4F mice. No travel effects were seen (Figure 4H). For genotype, it was found that male APP;E3F mice displayed lower triglyceride concentration than WT counterparts (Figure 4I, *p* = 0.0145).

### 2.4. GCRsim and Gamma Radiation Lowered Plasma Cytokine Levels in Female APP;E3F Mice

Plasma cytokine levels in GCRsim or gamma-irradiated APP;E3F and APP;E4F mice and sham-irradiated WT mice were quantified by ELISA to detect irradiation effects 7 months post-exposure (Figure 5A–X).

Irradiation with 0.75 Gy GCRsim had a modest effect on plasma IL-5 levels, with irradiated APP;E3F mice of both sexes showing a trend towards lower levels (Figure 5M, *p* = 0.0814 and *p* = 0.0534 in female and male, respectively). Next, radiation effects were observed for IL-6 in female APP;E3F mice (Figure 5P), with plasma levels being significantly lower in 0.75 Gy GCRsim-irradiated individuals (*p* = 0.0464) and showing a trend towards lower levels in 2 Gy gamma-irradiated mice (*p* = 0.986) compared to sham controls, respectively. For KC/GRO, plasma levels were significantly lower in 0.75 Gy GCRsim-irradiated animals compared to sham controls (Figure 5S, *p* = 0.0401 and *p* = 0.0495 in females and males, respectively). Furthermore, plasma levels of TNF-α were significantly decreased by 0.75 Gy GCRsim and 2 Gy gamma radiation in female APP;E3F mice compared to sham controls (Figure 5V, *p* = 0.0073 and *p* = 0.038). In contrast, plasma levels of IFN-γ, IL-10, IL-1β, and IL-2 were not affected by irradiation with 2 Gy gamma or 0.75 Gy GCRsim (Figure 5A,D,G,J).

Traveled, sham-irradiated male APP;E3F mice had significantly lower plasma levels of IL-10 (Figure 5E, *p* = 0.0392), IL-2 (Figure 5K *p* = 0.0392), and TNF-α (Figure 5W, *p* = 0.0381) compared to their non-traveled counterparts. Contrasting this, traveled, sham-irradiated male APP;E4F mice had significantly higher plasma IL-6 concentrations than their non-traveled controls (Figure 5Q, *p* = 0.0283). No effects of travel were found for plasma levels of IFN-γ, IL-1β, IL-5, and KC/GRO (Figure 5B,H,N,T).

Genotype influenced IL-1β plasma levels with male APP;E3F mice, showing a trend towards higher levels compared to WT counterparts (Figure 5I, *p* = 0.0567). A sex-dependent effect was seen for IL-5 levels, which were significantly higher in female APP;E3F mice compared to male counterparts (Figure 5O, *p* = 0.0303). Next, IL-6 levels were higher in the plasma of female APP;E3F mice compared to WT (*p* = 0.0057), APP;E4F (*p* = 0.0301), and male (*p* = 0.0778) counterparts (Figure 5R). Similarly, female APP;E3F mice displayed significantly higher levels of plasma TNF-α than APP;E4F (*p* = 0.0102) and male counterparts (*p* = 0.0479, Figure 5X). KC/GRO levels were significantly lower in female APP;E4F mice compared to WT (*p* = 0.0207) and showed a trend compared to APP;E3F (*p* = 0.0593, Figure 5M). No effects of sex or genotype were detected for plasma levels of IFN-γ, IL-10, and IL-2 (Figure 5C,F,L).

### 2.5. GCRsim and Gamma Irradiation Had Non-Significant Effects on ApoE and Aβx-42 Concentrations in the Brain

In brain homogenates of GCRsim or gamma-irradiated APP;E3F and APP;E4F mice, concentrations of human ApoE and Aβx-42 were quantified by use of an ELISA to detect effects of radiation 7 months post-exposure (Figure 6A–F).

For 2 Gy gamma radiation, a trend towards lower human ApoE levels was detected in male APP;E3F mice compared to sham controls (Figure 6A, *p* = 0.0856), while no effects of 0.75 Gy GCRsim irradiation were found. Travel decreased the human ApoE concentration significantly in female APP;E3F mice in comparison to their non-traveled counterparts (Figure 6B, *p* = 0.0419). While genotype had no effects on the human ApoE concentration, male APP;E4F mice showed significantly higher levels of brain human ApoE than their female counterparts (Figure 6C, *p* = 0.0813), and human ApoE levels were higher in female and male ApoE KI mice than WT mice, as expected (Figure 6C).

Aβx-42 levels were non-significantly increased by 0.75 Gy GCRsim radiation in female APP;E4F mice compared to sham controls (Figure 6D, *p* = 0.0612). No other radiation effects were detected. No travel effects were found. While no differences between APP;E3F and APP;E4F mice were seen, male APP;E4F mice showed a trend towards lower Aβx-42 levels in comparison to their female counterparts (Figure 6F, *p* = 0.0997).

### 2.6. Gamma- and GCRsim-Irradiated Mice Had Immediate and Long-Term Differences in Their Gut Bacterial Composition That Correlated to Alzheimer’s Disease Phenotypes

LEfSe analysis of differentially abundant bacterial taxa revealed that both gamma- and GCRsim-irradiated mice had differentially abundant bacteria in their stool compared to sham-treated mice at the amplicon sequence variant (ASV) level (Figure 7A–H) one day after irradiation. Differences were observed in both female and male mice and in mice with different genotypes with regard to ApoE. Among differentially abundant bacteria, we observed several bacterial species belonging to classes *Bacterodia* and *Clostridia*. For instance, female APP;E3F mice treated with gamma radiation had a higher abundance of several *Bacterodia* ASV belonging to the *Prevotellaceae* family and *Parabacteroides* genus (Figure 7A). Female mice APP;E3F treated with gamma radiation, on the other hand, had lower levels of an ASV belonging to the *Bacteroidia* genus *Muribaculum* and a higher abundance of an ASV assigned to the *Dubosiella* genus. We also observed ASVs that were different as long as 145 and 190 days after irradiation when comparing irradiated mice to sham mice (Figure 7I,J). To understand if these differences in bacterial composition could be linked to phenotypic differences observed in irradiated mice, long-term differentially abundant ASVs were correlated to measurements of brain amyloid plaque levels and behavioral outcomes from the forced-alternation Y-maze (Figure 7K). An interesting observation was that an ASV from the genus *Muribaculaceae* that was more abundant in GCRsim-irradiated male APP;E4F mice (compared to sham) had a positive correlation to ThioS-stained amyloid plaques in the same group (Spearman’s R = 0.48, *p* = 0.034) and that an ASV belonging to *C. stonquefichus* that was less abundant in gamma-irradiated male APP;E3F mice correlated negatively with the Y-maze test for spatial memory (Spearman’s R = −0.57, *p* = 0.0038). Figure 7L–O illustrate how these bacteria changed in abundance over time in male and female mice.

### 2.7. Genotype Influenced the Gut Microbial Composition in Control and Irradiated Mice

We observed that the genotype of the mice was the biggest contributor to beta diversity in the analyzed samples (Figure 8A). Indeed, among mice who travelled to the Brookhaven National Laboratory facility, all mice were clearly separated by genotype, regardless of radiation treatment at the facility (Figure 8B). Differences in ASVs could be observed 1 day after irradiation (Figure 8C), and there were persistent changes at later time points (Figure 8D).

## 3. Discussion

Our results in vivo in amyloid mice suggest that the types of radiation found in deep space may cause changes in behavior and pathology in astronauts later in life. Many studies have focused on the effects of HZE particles, such as ^56^Fe, at both high and low irradiation doses, with the latter being more relevant to spaceflight. However, a better representation of the radiation that astronauts traveling through deep space may experience is now available using mixed ion GCRsim instead of single-ion irradiation. However, little is known yet about the interaction of GCRsim radiation effects with age, genotype, and genetic predispositions to AD, as well as the AD pathology itself. In this study, we found sex-, genotype-, and travel-dependent late CNS effects of a simplified five-ion mixed-field beam and gamma rays on female and male AD-like KI mice on a human ApoE3 or ApoE4 background. A better understanding of the late CNS effects of deep space radiation on humans becomes more important, as NASA is planning manned missions into deep space, beyond low Earth orbit, within the next few years.

We observed genotype-, sex-, and travel-specific effects on behavior 6 months post-irradiation. No clear pattern was found for the effect of genotype. APP;E3F mice showed higher locomotive activity, while APP;E4F were more anxious. Female mice displayed more anxiety and stronger motor coordination than male mice, while males showed better motor learning, and outcomes for locomotive behavior were mixed. While no radiation-specific effects were found in gamma-irradiated animals, the behavior of male APP;E3F mice in the SNYM was negatively affected by GCRsim irradiation, compared to sham controls, suggesting that irradiation was detrimental for spatial working memory. A beneficial influence was observed for locomotion in male APP;E3F mice and motor coordination in female APP;E4F mice. Interestingly, a positive effect of travel was observed for anxiety-like behavior, locomotion, motor learning, and spatial memory in non-irradiated, traveled mice compared to non-traveled controls. While no effects of radiation or sex on brain pathology were detected 7 months post-irradiation, travel and genotype affected the outcome. While APP;E3F mice displayed more microhemorrhages, APP;E4F mice showed a higher fibrillar amyloid load in the HC. Remarkably, traveled controls harbored lower total Aβ loads compared to non-traveled mice, in line with the behavioral outcomes, possibly due to travel-related stress.

For locomotion in OF, we found no significant effects. In contrast to this, we have previously reported for APP/PS1 mice that 0.10 Gy ^56^Fe radiation increased locomotion in females and 0.50 Gy ^56^Fe radiation in males 1.5 months post-irradiation [25]. After 7 months post-irradiation, 0.50 Gy ^56^Fe radiation increased locomotive behavior in female APP/PS1 mice. Furthermore, female and male transgenic mice were hyperactive compared to WT mice [26]. While no influence of genotype was found here, other studies have reported less locomotion in female and male human ApoE4 targeted replacement (TR) mice compared to ApoE3 TR mice [54,55,56]. We found that anxiety-like behavior in OF was increased in female traveled and sham-irradiated APP;E4F mice compared to non-traveled controls. Female APP;E4F mice were more anxious than their male and APP;E3F counterparts, both after sham-irradiation. While one study reported higher anxiety levels in ApoE4 TR mice compared to ApoE3 [57], in another one, the opposite was seen [54], and others detected no significant difference between ApoE3 and ApoE4 TR mice [55,56]. No radiation-specific effects on anxiety were seen, in line with a previous study of our lab [25].

We found that 0.75 Gy GCRsim radiation increased motor coordination in rotarod tests for female APP;E4F mice compared to sham controls. We have previously reported that 0.50 Gy ^56^Fe radiation increased the average latency in male APP/PS1 mice [26], while contradictory outcomes were found for a lower dose of 0.10 Gy ^56^Fe radiation [25,26]. We further found that female APP;E3F mice had a better outcome than their male counterparts, while no genotype-specific effects were detected. Contrasting this, other studies have reported that female and male ApoE4 TR mice performed better than ApoE3 TR mice [54] and male WT mice better than transgenic APP/PS1 counterparts [26], all sham-irradiated. In this study, motor learning in the rotarod test was improved in female traveled APP;E3F mice compared to non-traveled controls, suggesting that travel might have a beneficial effect. Male WT mice showed more improvement on the rotarod task than females. In contrast to a previous study of our lab that reported a better outcome in female APP/PS1 mice six weeks after 0.50 Gy ^56^Fe radiation [25], we found no radiation-specific effects, in line with another study of our lab that focused on long-term effects [26], similar to this study.

We found no significant differences for long-term memory after 24h in the NOR task. However, in a different study that used APP/PS1 mice, it was reported that female mice showed cognitive impairments after 1 Gy ^56^Fe radiation and males after 0.10 and 1 Gy ^56^Fe radiation [24]. In line with this, female ApoE3 and ApoE4 TR mice displayed memory deficits after 1 Gy ^56^Fe radiation [55].

Behavior in the SNYM was affected by radiation, as seen in our study. First, irradiation with 0.75 Gy GCRsim increased the percentage distance that male APP;E3F mice traveled in the novel arm in the SNYM compared to sham controls, indicating that GCRsim promoted hyperactivity. Radiation effects on locomotion were shown in previous studies of our lab, which investigated the influence of ^56^Fe radiation, a component of GCR, on APP/PS1 mice. In comparison to sham controls, we found an increased level of locomotion in the Y-maze in male mice irradiated with 0.50 Gy, while female mice showed an increase only after 0.10 Gy [25], which is much lower than the GCRsim dose of 0.75 Gy that was used in this study. In contrast to this, another study found an increase in the traveled distance in OF for female mice irradiated with 0.5 Gy when testing behavior at a later time point [26]. However, in male mice, irradiation did not show significant effects on locomotion, or it even decreased the traveled distance in OF when comparing doses of 0.10 and 0.50 Gy to sham controls [26].

The SNYM can also be used to assess spatial memory by measuring the time spent in the novel arm. We found that 0.75 Gy GCRsim radiation impaired spatial short-term memory in male APP;E3F mice. In comparable studies, spatial memory was quantified by different tests. The spontaneous-alternation Y-maze task was utilized in previous studies of our lab. After 0.10 and 0.50 Gy doses of ^56^Fe radiation, APP/PS1 showed no significant differences compared to sham controls 1.5 months post-irradiation [25]. In contrast, male mice displayed memory impairments 7 months after irradiation [26], similar to the findings in our study. In another study, where male WT mice were irradiated with 1 or 2.5 Gy gamma or 0.10 or 1 Gy ^56^Fe, decreased spatial memory was seen for 1 Gy doses of both radiation types, as assessed by the T-maze 5 months post-irradiation. Interestingly, these differences disappeared when mice were retested 9 months post-irradiation [21], pointing towards the existence of repair mechanisms for radiation-induced changes. In line with this, male WT mice tested for spatial memory in a Radial Arm Water Maze (RAWM) between 9–11.5 months of age after exposure to 0.50 Gy 5-ion GCRsim at 6 months of age displayed significant deficits compared to sham controls [12]. Moreover, male WT mice displayed synaptic loss in the HC after irradiation with GCRsim [11]. In addition, it has been shown that irradiation decreases the rate of action potentials and causes an impairment of hippocampal LTP in WT mice, which negatively interferes with hippocampus-dependent memory [9,10]. We did not find genotype-dependent effects on spatial memory in this study, but differences between ApoE3 TR and ApoE4 TR mice were detected using the Morris Water Maze (MWM) task; while ApoE3 TR mice showed a decrease in memory compared to ApoE4 TR mice 3 months after 1 Gy ^56^Fe irradiation, no differences were reported for 2 Gy ^56^Fe radiation [55,56]. In contrast, the opposite was seen for 13 months after irradiation with 3 Gy ^56^Fe; ApoE4 TR mice displayed more deficits in spatial memory than ApoE3 TR mice [42].

While some radiation-specific effects of GCRsim were found in this study, there were no significant differences for gamma radiation on behavior and brain pathology. Indeed, GCR poses a greater risk to living organisms than gamma rays due to differences in mass and charge. Gamma rays are only sparsely ionizing, but HZE particles display an increase in ionization density as the charge increases [5]. Female WT mice were irradiated with 1.6 Gy ^56^Fe or 2 Gy of ^137^Cs gamma rays. For HZE ^56^Fe particles, they show higher oxidative stress in cortical cells, which leads to very high molecular damage, consisting of deoxyribonucleic acid (DNA) damage and cell death, and subsequently high neuronal injury and reactive gliosis. While the pathway is the same for sparsely ionizing gamma rays, the effects are less detrimental [20]. This DNA damage includes complex molecular machinery that governs the cell’s response to such extreme stress. Notably, Apoptosis-Inducing Factor (AIF) plays a critical role in this process. Upon exposure to HZE particles, AIF translocates from the mitochondria to the nucleus, where it forms a DNA–degradosome complex with proteins like Cyclophilin A (CypA) and histone H2AX. This complex facilitates large-scale DNA fragmentation through its nuclease activity, effectively degrading chromosomal DNA and leading to cell death [58]. Similar findings were made in rats exposed to ^56^Fe ions; a 3.2× higher frequency of chromosome aberrations was found compared to irradiation with ^60^Co gamma rays [59]. A comparison of 0.001–0.10 Gy ^4^He to 0.50–4 Gy ^137^Cs gamma rays also showed a stronger effect of the GCR component on the behavior of rats [60].

In this study, total Aβ load, assessed by hippocampal S97 Aβ staining, was not affected by radiation. In contrast to this, previous studies have shown for male APP/PS1 mice that 6E10 staining in the HC and frontal cortex (FC) was increased after 1 Gy ^56^Fe radiation compared to sham controls [24] and that hippocampal R1282 staining was increased after 0.50 Gy ^56^Fe radiation compared to 0.10 Gy [26]. However, in the 3xTg mouse model, no radiation-specific effects on 6E10 staining were found after doses of 0 Gy, 0.10 Gy, and 1 Gy ^56^Fe radiation [27]. No sex- and genotype-specific effects were detected here, while it was reported before that female mice showed a stronger plaque pathology than males [25,26,27].

In addition to total Aβ load, fibrillar amyloid was assessed in this study by hippocampal Thioflavin S staining. Amyloid fibrils are formed by the deposition of Aβ and act as a reservoir of soluble Aβ oligomers. In our study, we did not find significant differences between the sexes, even though a previous study of our lab reported a better outcome for male APP/PS1 mice compared to females [25]. We detected no influence of radiation on fibrillar amyloid levels, while it was shown in male APP/PS1 mice that 1 Gy ^56^Fe radiation increased the amount of Congo Red staining in the HC and FC [24]. Contrasting this finding, we reported previously that female APP;PS1 mice displayed less hippocampal Thioflavin S staining 2 months after irradiation with 0.10 and 0.50 Gy ^56^Fe radiation compared to sham controls [25]. However, when brain pathology was assessed after 8 months, no significant differences were found for fibrillar amyloid [26].

Insoluble Aβx-42 levels, as quantified by ELISA, showed a trend towards increased Aβx-42 in female APP;E4F mice irradiated with 0.75 Gy GCRsim compared to sham-irradiated counterparts. While one study reported a significant increase in insoluble Aβx-42 levels in male APP/PS1 mice after 1 Gy and 0.1 Gy ^56^Fe radiation [24], the majority of studies that investigated effects of ^56^Fe radiation in APP/PS1 [25,26] or 3xTg mice [27] did not find radiation-dependent changes in Aβx-42 levels. Furthermore, we detected a trend towards higher Aβx-42 levels in male APP;E4F mice compared to female counterparts, in line with previous studies [25,26].

We found that irradiation with 0.75 Gy GCRsim lowered plasma IL-5, IL-6, TNF-α, and KC/GRO, while 2 Gy gamma radiation lowered plasma IL-6 and TNF-α, both in female APP;E3F mice. For male mice, lesser effects were observed, with KG/GRO and IL-5 being lowered by 0.75 Gy GCRsim in APP;E4F and APP;E3F mice, respectively. While TNF-α, IL-5, and KC/GRO (murine IL-8) are known to be pro-inflammatory [61,62], IL-6 has pro- and anti-inflammatory properties [63]. An increase in serum TNF-α and IL-6 in the CNS and periphery was observed in AD patients [64,65,66]. While this is in contrast to our hypothesis that irradiation worsens AD pathology, it has to be kept in mind that plasma cytokines were assessed only 7 months after irradiation, not reflecting the cytokine levels shortly after exposure. In addition, it was shown previously that cytokine levels show a biphasic response to irradiation [67,68].

Earlier studies showed an association of genotype with different levels of plasma lipoproteins [69]. We detected higher levels of total plasma cholesterol in male APP;E3F mice compared to APP;E4F. The opposite was reported for APP/PS1 with a human ApoE3 or ApoE4 KI at 6–8 months of age [70]. In older mice of the same model, no significant differences between the genotypes were found at 8–16 [71] or 15 [72] months of age. We found no differences between APP;E3F and APP;E4F for HDL cholesterol and triglycerides, in line with a previous study of our lab using APP/PS1;E3 and APP/PS1;E4 mice at 15 months old [71].

A surprising finding of this study was that travel might have a positive effect. Travel decreased the total hippocampal Aβ load significantly in female APP;E3F mice, while male APP;E3F mice trended towards a lower plaque load. In addition, male APP;E3F mice showed a trend towards enhanced short-term memory. Furthermore, it significantly decreased ApoE levels in female APP;E4F mice. The transportation of animals to BNL and sham irradiation may have imposed acute stress. While it has been shown in several studies that acute and chronic stress exacerbates plaque load [73,74,75,76,77,78,79,80,81] and worsens cognition [73,74,82,83] in AD-like animal models [84], at least some positive effects of acute stress have been reported. Application of endoplasmic reticulum (ER) stress to different mammalian cells, including neurons, activates the ER chaperone binding immunoglobulin protein (BiP), which refolds misfolded proteins and thereby clears aggregates [85]. In addition, it was seen in rats that acute stress increased hippocampal neurogenesis and enhanced fear extinction memory [86]. Furthermore, acute stress rescued impaired LTP induction and synaptic plasticity in the hippocampal CA1 region of Tg2576 and 5XFAD mice [87]. Acute stress can also activate microglia in rodents via different receptors [88,89,90], leading to an increase in the pro-phagocytic marker CD68, the density of microglia, and microglial activation in the CA1 region in 3xTg-AD mice [91]. Activated microglia play an important role in Aβ degradation and clearance, as described by different publications [92,93,94]; thereby, they might be able to lower the Aβ load. Interestingly, a positive effect of travel was only observed for total, but not fibrillar, amyloid, even though microglia are able to engulf fibrillar amyloid [95,96] and degrade it when in an activated state [97]. It has been shown in vivo and in vitro that uptake of soluble and fibrillar amyloid occurs at comparable levels but is based on distinct mechanisms, including different intracellular distributions [98]. In addition, it was found in vitro that Aβ oligomers impair microglial phagocytosis and clearance of Aβ fibrils [99]. This mechanism might possibly have also interfered with the lower fibrillar amyloid clearance observed in this study. A beneficial effect of travel on AD pathology was not detected for APP;E4F mice. This could be due to ApoE4 being less efficient in advancing the proteolysis of soluble Aβ than the two other isoforms due to lower lipidation [100]. However, for non-traveled controls, some groups only included very low numbers of animals, hiding potential effects of stress. This was especially the case for APP;E4F mice (*n* = 2–3 per sex per group).

One mechanism by which cosmic radiation may change AD pathology is by modulating the gut microbial composition of the mice. The gut microbiome is intimately connected to the AD pathogenesis. Experimental models of AD that are raised in a germ-free environment have lower Aβ levels in the brain [41,101,102], suggesting that microbes can drive AD plaque deposition. We report both immediate and long-term changes in the gut microbiome of APP;E3F and APP;E4F mice, which also had distinct genotype-dependent microbial profiles. Interestingly, one amplicon sequence variant belonging to the genus *Muribaculaceae* was increased in male APP;E3F mice by GCRsim radiation and correlated to plaque burden. Depletion of strains of *Muribaculaceae* with the antibiotic metronidazole has previously been demonstrated to reduce plaques in an 5xFAD model of AD, linking this genus to AD pathology [103]. In humans, the abundance of a closely related bacteria from the *Bacteroidia* class, *Bacteroides*, has been shown to increase in aging and in AD [47,48,104].

## 4. Materials and Methods

### 4.1. Mice

This study was carried out in strict accordance with the recommendations in the Guide for the Care and Use of Laboratory Animals of the National Institutes of Health. Animal protocols were reviewed and approved by the Brigham and Women’s Hospital (BWH) (Protocol Number: 2016N000396) and BNL (Protocol Number: 504) Institutional Animal Care and Use Committees.

Mice used in this study were male and female C57BL/6J wildtype mice, APP^NL-F/NL-F^; ApoE4fl/fl mice, and APP^NL-F/NL-F^; ApoE3fl/fl mice that were bred and housed at BWH (Boston, MA, USA). APP^NL-F/NL-F^; ApoE4fl/fl and APP^NL-F/NL-F^; ApoE3fl/fl mice were generated by crossing of APP NL-F KI mice on a C57BL/6J background [105] to ApoE4/4 floxed [106] or ApoE3/3 floxed [106] mice on a C57BL/6NTac background, respectively. Dr. Takaomi Saido provided a material transfer agreement (MTA), which allowed for the crossing of this line with the targeted replacement human ApoE floxed mice. The Cure Alzheimer’s fund provided an MTA for ApoE4/4 floxed and ApoE3/3 floxed mice. Dr. David Holtzman (Washington University School of Medicine) provided breeding pairs of APP NL-F; ApoE4/4 floxed and APP NL-F; ApoE3/3 floxed mice.

All mice were always housed at a constant temperature on a 12 h light/dark cycle and had ad libitum access to food (PicoLab Rodent Diet #5053) and water. APP^NL-F/NL-F^; ApoE3fl/fl and APP^NL-F/NL-F^; ApoE4fl/fl mice (*n* = 8 mice per genotype and sex per group) and C57BL/6J WT mice (*n* = 4 mice per group) were irradiated at 14 months of age at BNL and underwent quarantine when they were returned to BWH. In addition, 15 transgenic mice (*n* = 3–4 mice per group) were not transferred to BNL or irradiated but still experienced the same quarantine period. Mice underwent behavioral testing at 20 months of age and were euthanized by CO_2_ exposure and tissues were harvested at 21 months of age. Of the 119 mice at the start of the study, 16 mice died or had to be euthanized during the study course, as indicated in Table 1. The 103 remaining mice were included in the data analysis. Deaths were evenly distributed between sexes, genotypes, and radiation types and doses. Final mouse numbers for each analysis can be found in the Appendix A.

### 4.2. Irradiation

Mice were transferred to the NASA Space Radiation Laboratory (NSRL) at BNL (Upton, NY, USA) for irradiation. They were allowed to acclimate at BNL for 3–7 days prior to subsequent relocation to NSRL for whole-body irradiation with a simplified 5-ion mixed-field beam (Table 2) at either 0 Gy (sham) or 0.75 Gy or with 2 Gy gamma radiation.

GCRsim used in this study consisted of 5 different ion species, which were delivered in rapid succession, in the order of protons, silicon-28, helium-4, oxygen-16, iron-56, and protons again. As displayed in Table 2, GCRsim irradiation included 1 GeV/n proton (0.2 KeV/micron) at an exposure of 0.2625 Gy, 600 MeV/n ^28^Si (50.4 KeV/micron) at an exposure of 0.0075 Gy, 250 MeV/n ^4^He (1.6 KeV/micron) at a dose of 0.135 Gy, 350 MeV/n ^16^O (20.9 KeV/micron) at an exposure of 0.045 Gy, 600 MeV/n ^56^Fe (173.8 KeV/micron) at an exposure of 0.0075 Gy, and 250 MeV/n proton (0.4 KeV/micron) at an exposure of 0.2925 Gy, totaling 0.75 Gy. Following a Poisson distribution, this equates to an estimated average of 735 million, 93,200, 53.5 million, 1.36 million, 27,500, and 462 million particle traversals per square cm, respectively.

The irradiation with gamma rays from a cesium-137 source consisted of a photon energy of 662 KeV (0.8 keV/micron) at 2 Gy at a rate of 112 mGy/minute. Following a Poisson distribution, this equates to an estimated average of 1.56 billion traversals per square cm.

For GCRsim irradiation, 2–3 non-anesthetized mice were placed into small plexiglass boxes (40 × 40 × 73 mm) with ventilation holes that were grid-like stacked within a 64-box frame. Boxes with males were on the bottom, and boxes with females were on the top. The sexes were separated by at least 3 rows of blank boxes. The frame containing the boxes was then carried into the “cave”, where mice were subjected to whole-body irradiation. All mice per radiation dose were irradiated at once. Afterwards, mice were removed from the cave, monitored for radioactive decay, and returned to their cages. Sham control mice underwent the same procedure but were not taken into the cave and irradiated but instead put into cages in the frame/holder for a period of time equivalent to that of their experimental counterparts.

For gamma ray exposures, mice were loaded individually into pre-shaped clean plexiglass containers with ventilation holes. These tubes were arranged in a foam holder and placed into an irradiator with a ^137^Cs source. Eight mice were irradiated at a time, and female and male mice were irradiated separately.

Three days after completion of all procedures, mice were returned to BWH, where they underwent quarantine before being re-introduced to the mouse colony. Quarantine included a fenbendazole diet for three weeks and two antibiotic treatments of selamectin, with one being at the beginning and one at the end of the quarantine period.

### 4.3. Behavioral Testing

#### 4.3.1. Open Field

This test assesses general health, locomotion, and anxiety-like behavior. A test arena, consisting of a square 39.2 cm × 40.2 cm base with blue-colored, non-transparent 30 cm high walls was used. Lights were kept dim and between each mouse, and the arena was cleaned with 10% ethanol. Mice were habituated to the room for at least 1 h before starting the experiment. The animals were allowed to explore the arena for 30 min. A computer-assisted infra-red tracking system (EthoVisionXT 14; Noldus, Wageningen, The Netherlands) was used to record the location of the animals in the arena and user-defined zone entries.

#### 4.3.2. Rotarod

This test was used to assess motor coordination and learning. The rotarod apparatus (Biological research rotarod for mice 47600 v04; Ugo Basile, Gemonio, VA, Italy) consists of a rod that is capable of rotating at different speeds either in a constant or accelerating mode. Animals were habituated to the room for at least 20 min. Mice were trained at the apparatus for 5 min at a constant speed of 4 rpm. If a mouse fell from the apparatus, it was recorded, and the mouse was placed back onto the rod until the 5 min session was completed. For testing sessions, the velocity accelerated from 4 rpm to 40 rpm over a period of 3 min. Animals were placed on the rod for a 5 min session, but if they fell down or turned with the rod for more than one round, the timer was stopped, and the mouse was placed back into the home cage. The test was repeated after 3 h to assess motor learning. Between each animal, the device was cleaned with 10% ethanol.

#### 4.3.3. Novel Object Recognition

This test was used to assess exploratory behavior and memory. The same arena of an open field was used. Every day, mice were habituated to the room for at least 20 min. Lights were kept dim, and between each mouse, the arena was cleaned with 10% ethanol. The first day, mice were acclimated to the arena for 10 min. Days 2–4 were used for training. Each day, mice were exposed to the arena containing two identical objects for 10 min per day. On day 5, mice were tested for 10 min by replacing one object by a novel object. Objects chosen were similar in size, and no preference for either object was detected in previous studies. Motion of mice was recorded via a computer-assisted infra-red tracking system (EthoVisionXT 14; Noldus, Wageningen, The Netherlands).

#### 4.3.4. Spatial Novelty Y-Maze

This test was used to assess spatial memory and locomotion. Mice were habituated to the room for at least 20 min before starting the experiment. The lights were dimmed, and the equipment was cleaned with 10% ethanol between each mouse and trial. Mice were placed in the start arm, which remained constant over all trials, and habituated to the Y-maze for 3 min. During this time, one arm, either left or right, was blocked off. For a 2 min inter-trial interval, mice were placed back into a holding cage. The blockade was removed for the 3 min test phase, and mice were placed back into the start arm. The acquisition period started when the mouse entered the left or right arm. Motion of mice was recorded via a computer-assisted infra-red tracking system (EthoVisionXT 14; Noldus, Wageningen, The Netherlands). The time spent in the novel arm was used to assess spatial memory, while distance traveled in the novel arm was used to assess locomotion.

### 4.4. Euthanasia and Tissue Harvest

Mice were euthanized by CO_2_ exposure, followed by exsanguination, bilateral thoracotomy, and removal of the brain. Blood was collected via cardiac punction before mice were transcardially perfused with 60 mL of 1× phosphate-buffered saline (PBS). Organs collected were the brain, heart, liver, kidneys, spleen, stomach, small intestine, and large intestine. The brain was divided sagittally, and the left hemisphere was fixed in 4% paraformaldehyde (PFA) in 1× PBS for 24 h at 4 °C and then transferred to a 0.082% sodium azide (Sigma-Aldrich, St. Louis, MO, USA) solution. Shortly before the sectioning, this hemisphere was cryoprotected with 10% sucrose for 12 h and 30% sucrose for 24 h. The right hemisphere was frozen using dry ice and stored at −80 °C for biochemical measurements.

Fecal samples were collected at four different time points. For baseline estimation, samples were collected 9 days before irradiation. To assess radiation effects, three fecal samples were collected 24h, ~5 months, and ~6 months post-irradiation. Fecal samples were stored at −20 °C until analysis.

### 4.5. Sectioning of Brains

The left brain hemisphere was sagittally sectioned at 30 µm on a sliding-knife microtome (HM 450; Epredia, Kalamazoo, MI, USA) with a −25 °C freezing stage. For this, 30% sucrose was used to build a platform and stabilize the hemisphere that was placed onto it. Sections were sequentially placed in 24-well plates, with one section per well. After the 24th section, section 25 was placed in well one, and the process was repeated. As a result, each well had ~5 equidistant sections, which were kept in 0.082% sodium azide (Sigma-Aldrich, St. Louis, MO, USA) solution and stored at 4 °C.

### 4.6. Staining of Brain Sections

#### 4.6.1. Staining Using S97 Polyclonal Antibody for General Aβ

For every mouse, on average, 5 equidistant, free-floating sections were stained. After washing with 1× PBS for 3 × 5 min, slices were incubated with 88% formic acid for 10 min and washed again (3 × 5 min dH_2_O; 1 × 5 min 1× PBS). For quenching, 3% H_2_O_2_, 10% methanol in 1× PBS was used for 15 min. Sections were transferred to 1× PBS with 0.1% Triton X-100 for 3 × 5 min. Blocking solution consisted of 10% Normal Goat Serum (#16210-064; Gibco^TM^, Waltham, MA, USA) and 0.1% Triton X-100 in 1× PBS and was applied for 1 h at RT while shaking. The pan-specific anti-Aβ sera S97 which was generated in rabbit (1:2000; a gift from Dr. Dominic Walsh, Brigham and Women’s Hospital) was diluted in 1× PBS with 5% Normal Goat Serum (#16210-064; Gibco^TM^, Waltham, MA, USA), and the sections incubated with the solution overnight at 4 °C on a shaker. The next day, sections were washed with 1× PBS for 3 × 5 min and incubated with a biotinylated goat-anti-rabbit antibody (1:2000; BA-1000; Vector Laboratories, Burlingame, CA, USA), diluted in 1× PBS with 5% Normal Goat Serum (#16210-064; Gibco^TM^, Waltham, MA, USA) for 30 min. After washing (1× PBS, 2 × 5 min), the VECTASTAIN^®^ Elite ABC-HRP Kit, Peroxidase (Standard) (PK-6100; Vector Laboratories, Burlingame, CA, USA), was used for 45 min as described by the manufacturer. Next, sections were washed with 1× PBS for 2 × 5 min and the 3,3′diaminobenzidine (DAB) substrate was added. The DAB Substrate Kit, Peroxidase (HRP) (SK-4100; Vector Laboratories, Burlingame, CA, USA), was used, and 2 drops of buffer (R1), 2 drops of DAB (R2), and 1 drop of H_2_O_2_ (R3) were added to 5 mL of dH_2_O. The staining was developed in this solution for 50 s and stopped by washing with dH_2_O for 2 × 5 min. Sections were washed with 1× PBS for 5 min and then mounted onto Fisherbrand Superfrost Plus Microscope Slides (Thermo Fisher, Waltham, MA, USA). After dehydrating the sections in a series of 50/70/95/100/100% EtOH (3 min each) and clearing lipids in two 3 min soaks in Histoclear (National diagnostics, Atlanta, GA, USA), slides were coverslipped using Permount (Thermo Fisher, Waltham, MA, USA).

#### 4.6.2. Thioflavin S Staining for Fibrillar Amyloid

One well, containing ideally 5 sections, was used per mouse. Brain sections were mounted onto Fisherbrand Superfrost Plus Microscope Slides (Thermo Fisher, Waltham, MA, USA). After the slices had dried, they were hydrated in dH_2_O for 3 × 2 min. Slices were incubated with filtered 1% aqueous thioflavin S (Sigma-Aldrich, St. Louis, MO, USA) for 8 min. Next, they underwent a series of 2 × 3 min 80% EtOH, 1 × 3 min 95% EtOH, and 3 × 2 min dH_2_O. Sections were coverslipped using polyvinyl alcohol (PVA)-1,4-diazabicyclo[2.2.2]octane (DABCO) coverslipping solution.

#### 4.6.3. Production of PVA-DABCO Coverslipping Solution

To obtain 25 mL of mounting medium, 6 g of glycerol and 2.4 g of PVA (Sigma-Aldrich, St. Louis, MO, USA) were mixed. A total of 6 mL of dH_2_O was added, and the solution was mixed overnight at RT. A total of 12 mL of 0.2 M Tris-HCl (pH 8–8.5; Bioland Scientific, Paramount, CA, USA) was added, and the mixture was heated to 50 °C under stirring. A total of 0.625 g of DABCO (Sigma-Aldrich, St. Louis, MO, USA) was mixed with the solution and centrifuged for 15 min at 5000× *g*. The supernatant could be used for coverslipping.

#### 4.6.4. Hemosiderin Staining (Modified Perl’s Prussian Blue) for Microhemorrhages

One well, containing approximately 5 sections, was used per mouse. Brain sections were mounted onto Fisherbrand Superfrost Plus Microscope Slides (Thermo Fisher, Waltham, MA, USA). After the slices had dried, they were hydrated in dH_2_O for 2 × 5 min. Next, slices were incubated with 2% K_4_FeCN_6_ (Sigma-Aldrich, St. Louis, MO, USA) and 2% HCl in dH_2_O for 30 min. Counterstaining was carried out using 50% nuclear fast red (Vector Laboratories, Burlingam, CA, USA) in dH_2_O for 5 min. For dehydration, slides underwent a series of 95% EtOH, 100% EtOH, and Histoclear (National diagnostics, Atlanta, GA, USA) for 2 × 2 min each. Coverslips were mounted onto slides using Permount (Thermo Fisher, Waltham, MA, USA).

### 4.7. Image Acquisition and Analysis

#### 4.7.1. Imaging S97 Antibody for General Aβ

Staining for general Aβ was imaged using an Eclipse E400 (Nikon, Minato, Tokyo, Japan) with a Nikon Plan Fluor 4× objective (Nikon, Minato, Tokyo, Japan) in combination with the camera software Gryphax V2.2.0. The percentage area positive for Aβ in the entire HC was quantified with a custom-written macro (by Dr. Praveen Bathini) for Fiji (Version 2.0.0, ImageJ (NIH)) after manually outlining the region of interest (ROI). Per animal, 2–3 consecutive sections, 720 μm apart from each other and in the middle plane of the hemibrain, were analyzed if possible. The threshold of detection was held constant during analyses, and the resulting percentage value for each animal was averaged across the sections.

#### 4.7.2. Imaging Thioflavin S for Fibrillar Amyloid

For imaging of Thioflavin S staining, an AxioImager A1 and a Zeiss Plan-apochromat 10× objective (Zeiss, Oberkochen, Germany) were combined with the software ZEN 3.4 (ZEN lite) (Zeiss, Oberkochen, Germany). Exposure time was kept constant over all sections at 330 ms, and intensity was set to 70%. Per animal, 2–3 consecutive sections, 720 μm apart from each other and in the middle plane of the hemibrain, were analyzed if possible. In the snapped pictures, the fluorescence signal was quantified in the HC with a custom-written macro (by Dr. Praveen Bathini) for Fiji (Version 2.0.0, ImageJ (NIH)) after the ROI was outlined manually. The threshold was kept constant across all sections. For each animal, the resulting percentage value was averaged across the sections.

#### 4.7.3. Imaging Hemosiderin Staining (Modified Perl’s Prussian Blue) for Microhemorrhages

Hemosiderin staining was analyzed using an Eclipse E400 (Nikon, Minato, Tokyo, Japan) with a Nikon Plan Fluor 10× objective (Nikon, Minato, Tokyo, Japan) in combination with the camera software Gryphax V2.2.0. Hemosiderin-positive dots associated with blood vessels were manually counted over the whole section. Per animal, ideally, three sections (three planes) were examined, and the number of dots was averaged across the sections.

### 4.8. Cholesterol/Triglyceride ELISA

Fujifilm LabAssay™ kits for Cholesterol E (635-50981; Fujifilm Wako Pure Chemical Corporation, Chuo-Ku, Osaka, Japan), HDL-Cholesterol E (299-96501; Fujifilm Wako Pure Chemical Corporation, Chuo-Ku, Osaka, Japan), and Triglyceride (632-50991; Fujifilm Wako Pure Chemical Corporation, Chuo-Ku, Osaka, Japan) were used to quantify total cholesterol, HDL cholesterol, and triglycerides, respectively. The assay kits were conducted following the procedures given by the kits’ inserts.

### 4.9. MSD Cytokine ELISA

A Proinflammatory Panel 1 (mouse) V-PLEX ELISA kit (Meso Scale Diagnostics, Rockville, MD, USA) was used to quantify plasma cytokine levels of IFN-γ, IL-1β, IL-2, IL-4, IL-5, IL-6, KC-GRO, IL-10, IL-12p70, and TNF-α. Plasma samples were diluted 1:2 with the diluent provided by the kit, and the ELISA was conducted following the procedures given by the kit insert. Levels of IL-12p70 and IL-4 were below the level of detection of the assay.

### 4.10. ApoE Sandwich ELISA

The brain homogenization protocols performed using this kit were conducted as previously described in Liu et al., 2019 [24]. The levels of ApoE protein were measured in guanidine-soluble brain homogenate samples diluted 1:10 using a sandwich ELISA, in which the antibodies HJ15.6 and HJ15.4B were used for capture and detection, respectively (a gift from Dr. David M. Holtzman, Washington University School of Medicine, St. Louis, MO, USA). Briefly, a 96-well PerkinElmer^TM^ ViewPlate^TM^ (PerkinElmer, Waltham, MA, USA) was coated with 50 µL of 10 µg/mL HJ15.6 antibody in a carbonate buffer (0.2 g NaN_3_, 2.93 g NaHCO_3_, 1.71 g Na_2_CO_3_ in 1 L of Milli-Q water at 9.6 pH), and the plate was then stored at 4 °C overnight. The plate was then washed 4 times with 190 µL/well of 1× PBS. Subsequently, the plate was blocked with 2% BSA in 1× PBS and incubated at 37 °C for 1h. During incubation, standards and 1:10 guanidine samples using an “ApoE buffer” (0.5% BSA, 0.025% Tween-20 in 1× PBS) were prepared. Immediately after incubation, the plate was once again washed 4 times with 190 µL/well of 1× PBS. The standards and samples were then loaded, and the plate was covered and stored at 4 °C overnight. The plate was again washed 4 times with 190 µL/well of 1× PBS, and 50 µL/well of 150 ng/mL of HJ15.4B antibody in ApoE buffer was added. The plate was then incubated at 37 °C for 1.5 h. Post-incubation, the plate was washed 4 times using 190 µL/well of 1× PBS, and 50 µL/well of 1:10,000 strep HRP 40 was added. The plate was then incubated for 1.5 h while shaking at room temperature. One last set of washing the plate with 190 µL/well of 1× PBS was conducted. A total of 50 µL/well of Super Slow ELISA TMB (Sigma-Aldrich, St. Louis, MO, USA) was added, and the plate was read using a Benchmark Plus microplate reader (Bio-Rad Laboratories, Hercules, CA, USA) at 650 nm at 5 min, 10 min, and 15 min after the Super Slow ELISA TMB was added.

### 4.11. MSD Multiplex Aβ Triplex-38/40/42 Protein ELISA

The brain homogenizations and ELISA protocol performed using this kit were conducted as previously described in Liu et al., 2019 [24]. Briefly, guanidine samples were mixed overnight at 4 °C and were centrifuged at 175,000× *g*’s for 60 min at 4 °C. The supernatant (insoluble fraction) was transferred, aliquoted, and stored at −80 °C. Cerebral levels of Aβx-38, x-40, and x-42 were measured simultaneously by a Human/Rodent 4G8 Aβ Triplex Ultra-Sensitive Assay (Meso Scale Diagnostics, Rockville, MD, USA). Levels of Aβx-40 and Aβx-38 were below the level of detection of the assay.

### 4.12. 16S Gut Microbiota Sequencing and Analysis

Fecal pellets were collected both at baseline (prior to travel) and at various timepoints throughout the experiment. Upon collection, the samples were promptly stored at −80 °C to preserve their integrity. DNA was extracted using a PowerLyzer DNA Extraction Kit (12855; Qiagen, Hilden, Germany) according to the manufacturer’s instructions. The 16S rRNA gene V45 region was amplified through PCR using HotMaster Taq DNA Polymerase and Hotmastermix (QuantaBio, Beverly, MA, USA), along with barcoded fusion primers (515F,926R) specifically designed by the Earth Microbiome Project (PMID: 22402401, 27822518). These primers were targeted to the V45 region and contained adaptors for MiSeq sequencing as well as single-index barcodes that allowed for the pooling of PCR products. Post-amplification, the DNA was quantified utilizing the Quant-iT PicoGreen dsDNA Assay Kit (P11496; Thermo Fisher Scientific, Waltham, MA, USA). To ensure equal representation of the samples, they were pooled in equimolar amounts of 100 ng per sample. Any excess primers were removed using a QIAquick PCR Purification Kit (28104; Qiagen, Hilden, Germany). Following the clean-up step, DNA quantification was repeated using a Qubit high sensitivity dsDNA Assay (Life Technologies, Waltham, MA, USA) to determine the final concentration. The DNA samples were combined at equal concentrations prior to submission for paired-end 300 bp read sequencing on the Illumina MiSeq platform at the Harvard Medical School Biopolymers Facility (Illumina, San Diego, CA, USA). To enhance the signal from a low-diversity library, 40% PhiX was spiked into the amplicon pool during sequencing. Data analysis was performed in Qiime2 v.2022.8, and the analysis was an adaption of the Moving Picture pipeline. Sequences were trimmed by 5 bases and truncated at 240 and 160 bases for forward and reverse sequences, respectively. Taxa were assigned using a classifier trained on the Silva 138 SSURef NR99 database [107]. Amplicon sequence variants (ASVs) were included in the analysis that were present in at least 2 samples, had at least 10 sequences across all samples, and were not considered contaminants (mitochondria, chloroplasts, Halomonas, Mycoplasma weyonii, and Candidatus Mycoplasma). Samples were included if they had at least 1000 sequences (4 out of 433 samples were excluded from the analysis based on these criteria). Differential abundance analysis was performed using LEfSe (linear discriminant analysis effect size) [108].

### 4.13. Statistical Analysis

Prism version 10.2.3 (Graphpad, San Diego, CA, USA) was used for data analysis. All data are expressed as the mean ± SEM. A value of *p* < 0.05 was considered significant (* *p* ≤ 0.05; ** *p* ≤ 0.01; *** *p* ≤ 0.001) and # *p* < 0.1 was considered a notable trend for all statistical tests. Data were analyzed using 3-way ANOVAs for sex, genotype, and dose/radiation type (i.e., F vs. M, E3F vs. E4F, 0 Gy vs. 0.75 GCRsim and 2 Gy gamma) and travel (i.e., F vs. M, E3F vs. E4F, no-travel vs. traveled sham). For behavior and hemosiderin data, 2-way ANOVAs for sex and genotype (i.e., F vs. M, WT vs. E3F vs. E4F) were applied to comparisons within sham-irradiated mice. The results of the ANOVAs were used to determine the *p* value between genotypes of the same sex, with unpaired *t*-tests being used between F and M mice of the same genotype. In cases of highly varied variances, tests were dropped to either a one-way ANOVA or *t*-test to lower the likelihood of a false-positive significance. For ANOVAs, the post hoc test applied was Fisher’s least significant difference (LSD). Likewise, the unpaired *t*-tests were not adjusted for multiple comparisons. It was advised by a Harvard Catalyst biostatistician for this study to keep as many mice as possible in each analysis due to the low *n* value of each group. Due to a couple of cases of extreme outliers, outcomes that had a z-score above 4.265 (1 in 100,000 chance) when accounting for all mice, in addition to a z-score above 2 within its own group, were excluded, which only occurred within the female APP;E3F cytokine results. Correlations were calculated with Spearman’s correlation.

## 5. Conclusions

In conclusion, this study is, to our knowledge, the first to investigate the long-term effects of low-dose GCRsim radiation and gamma rays on behavior and Alzheimer’s brain pathology and the gut microbiome in aged female and male APP;E3F, APP;E4F, and WT mice. We found that 0.75 Gy GCRsim radiation led to better motor coordination in female APP;E4F mice and higher locomotor activity in male APP;E3F mice but disturbed spatial short-term memory in male APP;E3F mice 6 months post-irradiation, while no overall effect of gamma radiation was detected. Surprisingly, we found a beneficial effect of travel on spatial short-term memory in male APP;E3F mice and total Aβ load in female and male APP;E3F mice. While this study might help to better protect astronauts traveling through outer space, additional studies are required to better understand the interaction between variants of the disease-associated risk gene ApoE and radiation.

## Figures and Tables

**Figure 1 ijms-25-09379-f001:**
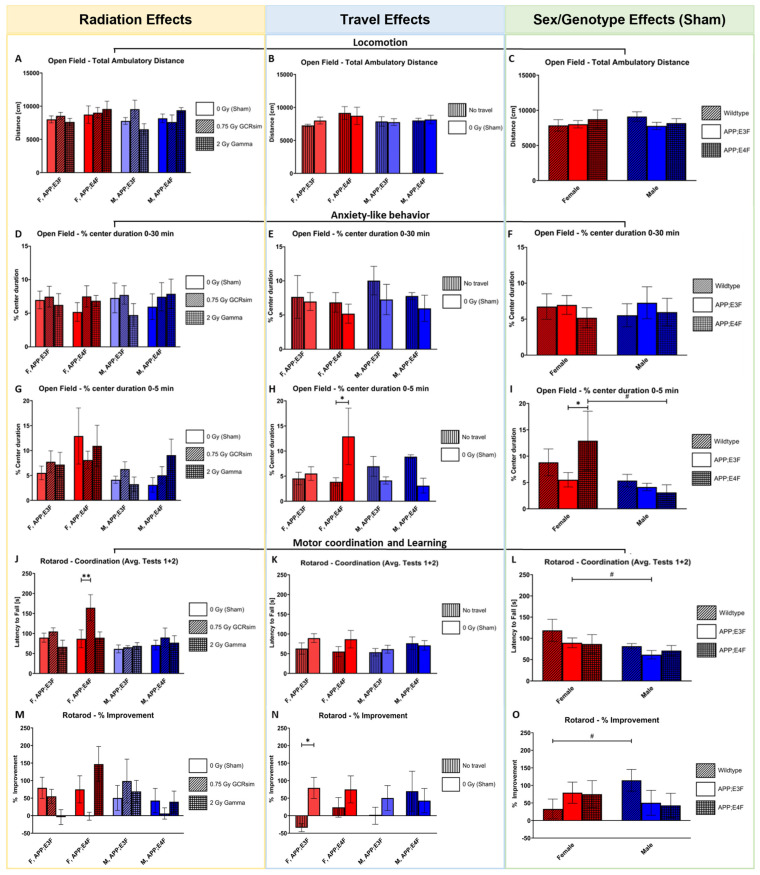
Performance of APP;E3F, APP;E4F, and wildtype (WT) mice in open field (OF) test was affected by travel, sex, and genotype, while in rotarod test, it was influenced by irradiation, travel, and genotype. Fourteen-month-old female and male APP;E3F and APP;E4F were irradiated with 0 Gy, 0.75 Gy simulated galactic cosmic radiation (GCRsim), or 2 Gy gamma at Brookhaven National Laboratory (BNL). WT mice were only sham-irradiated and traveled to BNL. All mice underwent behavioral testing 6 months post-irradiation, together with age-matched transgenic, non-traveled controls. (**A**–**C**) Effects of radiation, travel, sex, or genotype on locomotor activity, measured as total ambulatory distance in the OF, were analyzed. (**D**–**F**) Anxiety-like behavior was quantified as percentage time spent in the center of the OF for the full duration from 0–30 min. (**G**–**I**) Anxiety-like behavior as percentage center duration between 0–5 min. (**J**–**L**) Motor coordination was measured as latency to fall on the rotarod. (**M**–**O**) Percentage improvement between two rotarod trials was used to assess motor learning. #: *p* < 0.1, *: *p* < 0.05, **: *p* < 0.01. Data (**A**,**B**,**D**,**E**,**G**,**H**,**J**,**K,M**,**N**) were analyzed using three-way ANOVAs with Fisher’s least significant difference (LSD). Data (**C**,**F**,**I**,**L**,**O**) were analyzed using two-way ANOVAs with Fisher’s LSD to determine significance within sex and unpaired *t*-tests between sex. *p* values were not adjusted for multiple comparisons.

**Figure 2 ijms-25-09379-f002:**
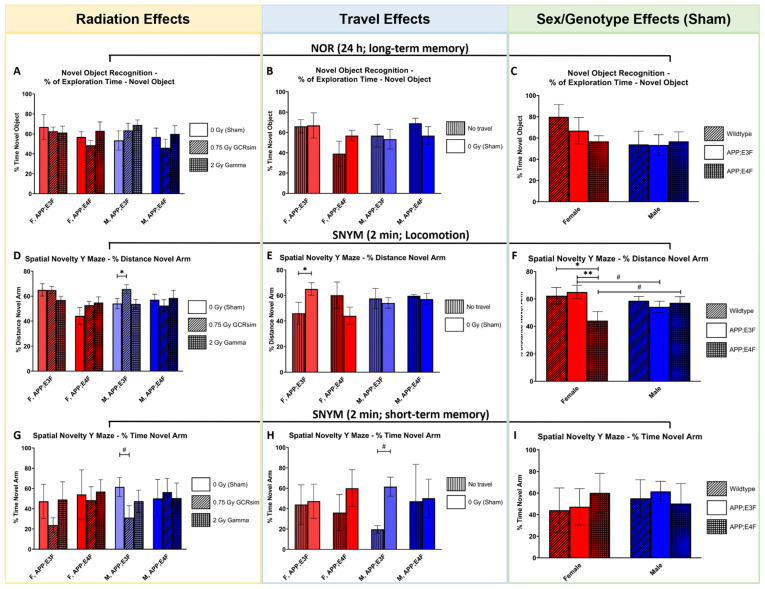
Performance of APP;E3F, APP;E4F, and WT mice in spatial novelty Y-maze (SNYM) test was influenced by radiation, travel, and genotype, while no effects on cognition in novel object recognition (NOR) test were detected. Fourteen-month-old female and male APP;E3F and APP;E4F were irradiated with 0 Gy, 0.75 Gy GCRsim, or 2 Gy gamma at BNL. WT mice were only sham-irradiated and traveled to BNL. All mice underwent behavioral testing 6 months post-irradiation, together with age-matched transgenic, non-traveled controls. (**A**–**C**) Long-term memory after 24h was measured with NOR (% time spent with novel object). (**D**–**F**) Locomotion was quantified as % distance in the novel arm of the SNYM. (**G**–**H**) Short-term memory after 2 min was analyzed as % time spent in the novel arm of the SNYM. #: *p* < 0.1, *: *p* < 0.05, **: *p* < 0.01. Data (**A**,**B**,**D**,**E**,**G**,**H**) were analyzed using three-way ANOVAs with Fisher’s LSD. Data (**C**,**F**,**I**) were analyzed using two-way ANOVAs with Fisher’s LSD to determine significance within sex and unpaired *t*-tests between sex.

**Figure 3 ijms-25-09379-f003:**
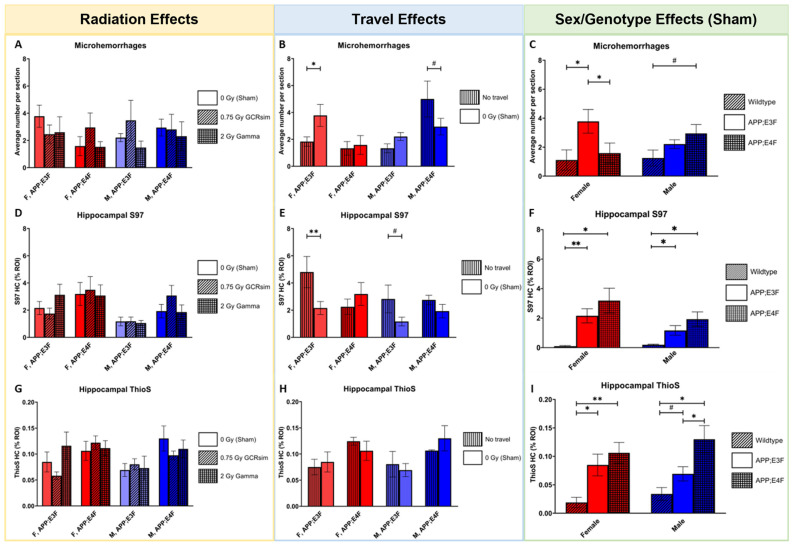
GCRsim and gamma radiation had no effects on microhemorrhages and Aβ levels in APP;E3F and APP;E4F mice 7 months post-irradiation. (**A**–**C**) Microhemorrhages were quantified based on Prussian blue staining for hemosiderin deposits across the whole-hemibrain section. (**D**–**F**) Total amyloid beta load was assessed by S97 Aβ immunohistochemical staining and quantified by the % region of interest (ROI) in the hippocampus (HC) using ImageJ (**G**–**I**). Fibrillar amyloid load was examined by Thioflavin S staining and quantified as the % ROI in the HC via ImageJ. #: *p* < 0.1, *: *p* < 0.05, **: *p* < 0.01. Data (**A**,**B**,**D**,**E**,**G**,**H**) were analyzed using three-way ANOVAs with Fisher’s LSD. Data (**C**,**F**,**I**) were analyzed using two-way ANOVAs with Fisher’s LSD to determine significance within sex and unpaired *t*-tests between sex. *p* values were not adjusted for multiple comparisons.

**Figure 4 ijms-25-09379-f004:**
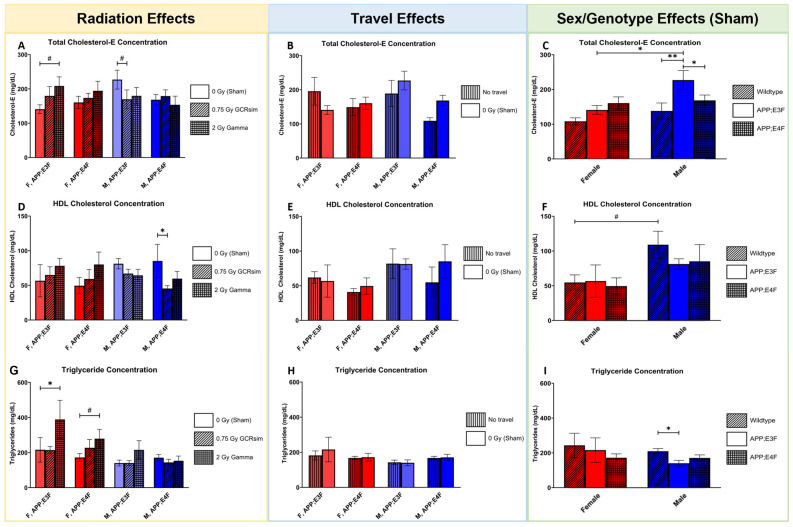
GCRsim and gamma radiation had effects on cholesterol and triglyceride levels in the plasma of APP;E3F and APP;E4F mice 7 months post-irradiation. (**A**–**C**) Total cholesterol concentration in the plasma was quantified using a Fujifilm Cholesterol E assay kit. (**D**–**F**) High-density lipoprotein (HDL) cholesterol concentration in the plasma was assessed via a Fujifilm HDL-Cholesterol E assay kit. (**G**–**I**) Triglyceride concentration in the plasma was determined by the use of a Fujifilm LabAssay Triglyceride assay kit. #: *p* < 0.1, *: *p* < 0.05, **: *p* < 0.01. Data (**A**,**B**,**D**,**E**,**G**,**H**) were analyzed using three-way ANOVAs. Data (**C**,**F**,**I**) were analyzed using two-way ANOVAs for within-sex comparisons and multiple *t*-tests within genotypes. *p* values were not adjusted for multiple comparisons.

**Figure 5 ijms-25-09379-f005:**
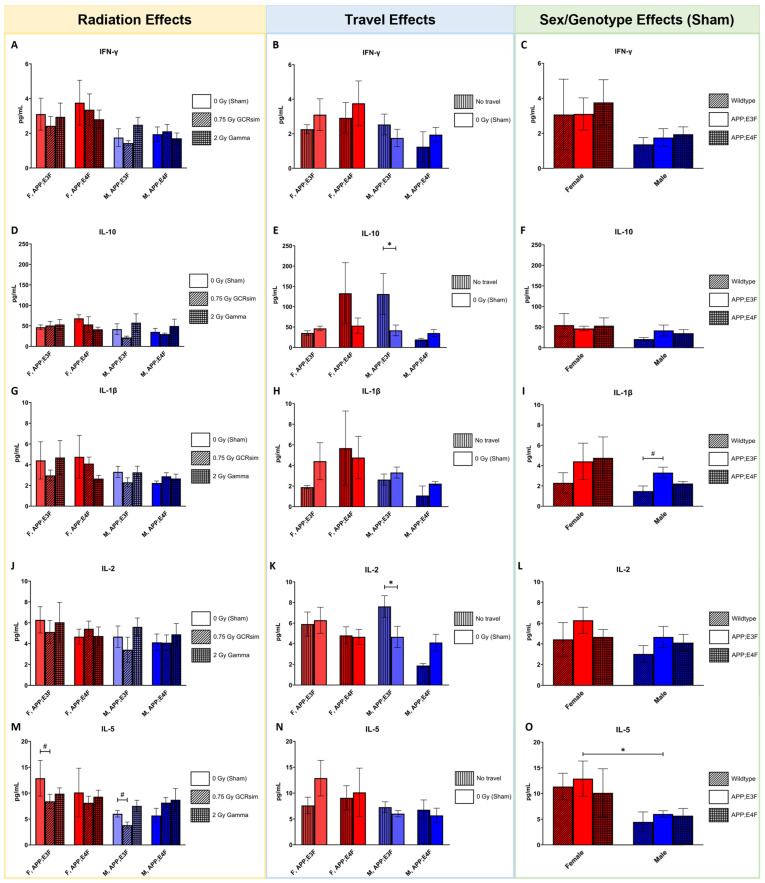
GCRsim and gamma irradiation decreased plasma cytokine levels in APP;E3F and APP;E4F mice 7 months post-irradiation. (**A**–**X**) Cytokine levels in the plasma were quantified using an MSD V-PLEX Proinflammatory Panel 1 ELISA. #: *p* < 0.1, *: *p* < 0.05, **: *p* < 0.01. Data (**A**,**B**,**D**,**E**,**G**,**H**,**J**,**K**,**M**,**N**,**P**,**Q**,**S**,**T**,**V**,**W**) were analyzed using three-way ANOVAs. Data (**C**,**F**,**I**,**L**,**O**,**R**,**U**,**X**) were analyzed using two-way ANOVAs for within-sex comparisons and multiple *t*-tests within genotypes. Data with highly different variances were compared with either one-way ANOVA or multiple *t*-test. *p* values were not adjusted for multiple comparisons.

**Figure 6 ijms-25-09379-f006:**
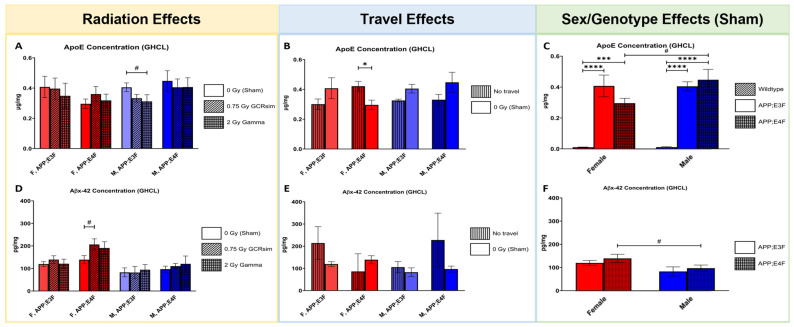
Effects of radiation, travel, sex, and genotype on ApoE and Aβx-42 levels in brain homogenates of GCRsim- and gamma-irradiated APP;E3F and APP;E4F mice 7 months post-irradiation. (**A**–**C**) ApoE concentration in the brain was quantified using an in-house sandwich ELISA. (**D**–**F**) Levels of Aβx-42 in the brain were quantified using an ELISA. #: *p* < 0.1, *: *p* < 0.05, ***: *p* < 0.001, ****: *p* < 0.0001. Data (**A**,**B**,**D**,**E**) were analyzed using a three-way ANOVA. Data (**C**,**F**) were analyzed using a two-way ANOVA for comparisons between sex and multiple *t*-tests within genotypes. Data with highly different variances were compared with either one-way ANOVA or multiple *t*-test. *p* values were not adjusted for multiple comparisons.

**Figure 7 ijms-25-09379-f007:**
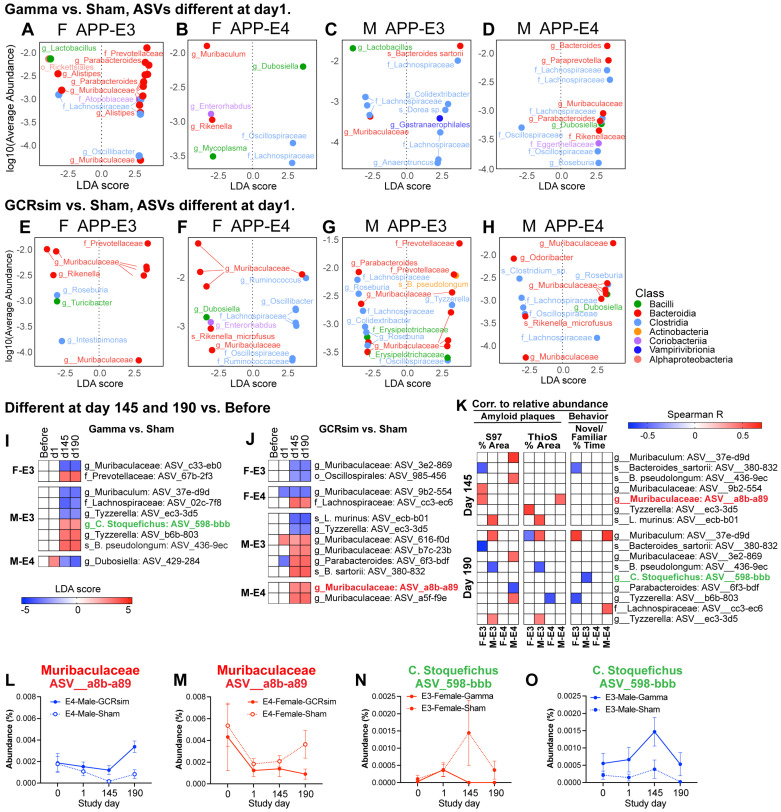
Gamma- and GCRsim-irradiated mice had immediate and long-term differences in their gut bacterial composition that correlated to Alzheimer’s disease phenotypes. Male (M) and female (F) APP-ApoE3 knock-in (E3) and APP-ApoE4 knock-in (E4) mice were transported (Tr) to the Brookhaven National Laboratory facility at 14 months of age and were exposed to either 0.75 GCRsim, 2 Gy gamma, or 0 Gy (Sham) radiation. Fecal pellets from the mice were collected before (0 days), the day after, 145 days after, and 190 days after irradiation and analyzed with 16S rRNA sequencing. Differences in the relative abundance of bacteria were analyzed with linear discriminant analysis (LDA) effect size (LEfSe). Several amplicon sequence variants (ASVs) were differently abundant one day after irradiation compared to sham controls. The graph shows each bacterial taxa that were significantly different as their LDA score (higher than 0 is more abundant, lower is less abundant) plotted against the logarithm of their average relative abundance (**A**–**H**). Heatmaps of bacteria that were significantly different from baseline at both 145 days and 190 days post-irradiation. All colored boxes had a *p* < 0.05 (**I**,**J**). Heatmap representing the Spearman R of relative abundance correlated with S97 and Thioflavin S (ThioS) staining of amyloid plaques and % of time spent in the novel arm of the forced-alternation Y-maze. All colored boxes had a *p* < 0.05 (**K**). The relative abundance over time of ASV a8b0bb4465364446253d4efd0ec16a89 (*Muribaculaceae*), which had a significantly higher abundance in male E4 mice and correlated with higher plaque burden in the same group (**L**,**M**). The relative abundance over time of ASV 598b197a5e45205c84efc7cb64258bbb (*Candidatus stoquefichus*), which had a significantly higher abundance in male E3 mice and correlated with higher plaque burden in the same group (**N**,**O**).

**Figure 8 ijms-25-09379-f008:**
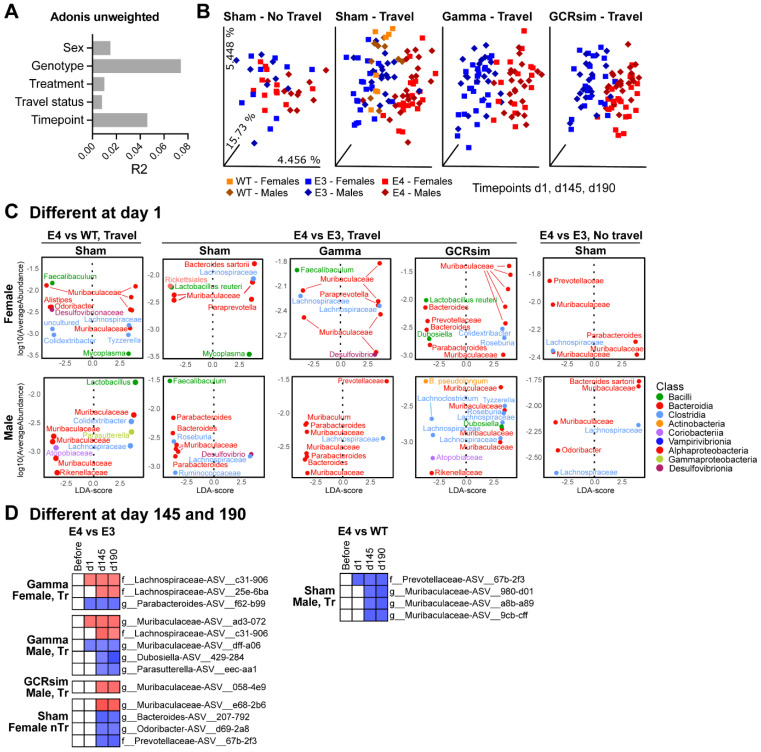
Genotype influenced the gut microbial composition in control and irradiated mice. Male (M) and female (F) APP-ApoE3 knock-in (E3) and APP-ApoE4 knock-in (E4) mice travelled (Tr) to the Brookhaven National Laboratory facility at 14 months of age and were exposed to either 0.75 GCRsim, 2 Gy gamma, or 0 Gy (Sham) radiation. Fecal pellets from the mice were collected before (0 days), the day after, 145 days after, and 190 days after irradiation and analyzed with 16S rRNA sequencing. Bacterial taxa that were detected in at least two-thirds of the samples were included in the analysis. Differences in the relative abundance of bacteria were analyzed with linear discriminant analysis (LDA) effect size (LEfSe) at the amplicon sequence variant (ASV) level. Unweighted permutational analysis of variance (Adonis test) of sex, genotype, treatment, travel status, and timepoint of the microbiome samples (**A**). Unweighted UniFrac distance colored by genotype and split by type of radiation (**B**). Differently abundant ASVs one day after irradiation compared to sham controls. The graph shows each bacterial taxa that were significantly different as their LDA score (higher than 0 is more abundant, lower is less abundant) plotted against the logarithm of their average relative abundance (**C**). Heatmaps of bacteria that were significantly different from baseline at both 145 days and 190 days post-irradiation. All colored boxes had a *p* < 0.05 (**D**).

**Table 1 ijms-25-09379-t001:** Number of mice used in this study per group/dose/sex.

Genotype/Sex	Sham	0.75 Gy GCRsim	2 Gy Gamma	No Travel	Total
APP^NL-F/NL-F^; ApoE3fl/fl female	6/8	8/8	6/8	4/4	24/28
APP^NL-F/NL-F^; ApoE3fl/fl male	8/8	8/8	7/8	4/4	27/28
APP^NL-F/NL-F^; ApoE4fl/fl female	5/8	7/8	8/8	3/4	23/28
APPN^L-F/NL-F^; ApoE4fl/fl male	6/8	7/8	7/8	2/3	22/27
C57BL/6J wildtype female	4/4	-	-	-	4/4
C57BL/6J wildtype male	4/4	-	-	-	4/4
Total	33/40	30/32	28/32	13/15	103/119

**Table 2 ijms-25-09379-t002:** Composition of the simplified 5-ion mixed-field beam normalized to 500 mGy.

Ion Species	Energy (MeV/n)	LET	Dose (mGy)	Dose Fraction	Delivery Order	Hits	Dose Ratios	
Proton	1000	0.2	174.1	0.35	1	537.1	Proton	0.74
^28^Si	600	50.4	5.7	0.01	2	0.078	Alpha	0.18
^4^He	250	1.6	90.2	0.18	3	39.2	HZE	0.08
^16^O	350	20.9	29.1	0.06	4	0.95		
^56^Fe	600	173.8	5.1	0.01	5	0.020		
Proton	250	0.4	195.9	0.39	6	340.2		

## Data Availability

The datasets generated and/or analyzed during the present study are available from the corresponding author upon reasonable request.

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
