# Peer review of "Cognitive Effects of Simulated Galactic Cosmic Radiation Are Mediated by ApoE Status, Sex, and Environment in APP Knock-In Mice"

_ijms, 2024, doi:10.3390/ijms25179379_

Round 1
Reviewer 1 Report
Comments and Suggestions for Authors
The current study investigates the effects of simulated galactic cosmic radiation on memory in aging mice. The study used AppNL-F/NL-F knock-in mouse model, in which the brain exhibit AD-like pathology. The study should be considered novel and correctly performed. Minor errors do not affect my high evaluation of the work.
Line 32 - please define what CNS stands for.
Line 97 - the first use of the abbreviation AD is in line 94.
Line 143 - similarly, the first “knock-in” term appears in line 138 and there it should be shortened to KI.
Line 152 - please define more precisely what the authors' research hypothesis is.
Line 715, 717 - in some cases the number n = 3 or 4. Did the authors do a power test to justify such small research groups (in my opinion too small to get reliable statistical results).
Line 812 - anatomically the term “gastrointestinal block” does not exist. Please provide the correct names of the organs harvested
Line 838 - the term “anti-Aβ anti-serum” sounds ridiculous. Either “antibodies raised against Aβ” or “ anti-Aβ sera”.
Line 838 - how did the authors check the specificity of this antibody?
Line 838 - please attach any photographs (documentation) of the stains performed.
Line 989 - did the authors check the normal distribution at all? With what test?
Reviewer 2 Report
Comments and Suggestions for Authors
The manuscript titled “Cognitive effects of simulated galactic cosmic radiation are mediated by ApoE status, sex, and environment in APP knock-in mice” by Wieg, L.; et al. is a scientific work where the authors study the effect of gamma rays and simulated galactic cosmic radiation in APP knock-in model mice at different irradiation doses. The most relevant outcomes found revealed the negative effect of this class of irradiation and the onset of neurodegenerative diseases and alterations in the microbiome in the examined specimens. This topic is interesting and the manuscript is generally well-written. However, it exists some points that need to be addressed (please, see them below detailed point-by-point) to improve the scientific quality of the submitted manuscript paper before this article will be consider for its publication in the International Journal of Molecular Sciences.
1) Keyword. The authors should consider to add the term “APP mouse models” in the keyword list.
2) “There is an increasing demand for deep space exploration (…) astronauts will be exposed (…) Galactic Cosmic Radiation (…) (HZE) particles” (lines 50-56). Here, even if I agree with this statement provided by the authors it should not be neglected the potential impact of galactic cosmic radiation on the crew of commercial flights [1]. Some quantitative insights should be furnished in this regard.
[1] Toprani, S.M.; et al. Cosmic Ionizing Radiation: A DNA Damaging Agent That May Underly Excess Cancer in Flight Crews. Int. J. Mol. Sci. 2024, 25, 7670. https://doi.org/10.3390/ijms25147670.
3) Results. Why did the authors only test the effect of 0.75 Gy and 2.00 Gy radiation (in addition to no radiation experiments) and not a broad range of values? (e.g. lower galactic cosmic radiation values around 0.2-0.3 Gy are received those astronauts who work at the International Space Station, among other situations). Some discussion should be provided in this regard.
4) Then, this research examined the effect of gamma radiation but not X-ray radiation on the wild-type and APP knock-in mouse models. Could this aspect negatively interfere in the data interpretation?
5) “For the HZE particle they show higher oxidative stress in cortical cells (…) deoxyribonucleic acid (DNA) damage and cell death (…) reactive gliosis” (lines 585-588). Here, even if I also agree with this information stated by the authors it should be detailed the DNA molecular machinary which leads to DNA degradation inner the cells [2].
[2] Novo, N.; et al. Beyond a platform protein for the degradosome assembly: The Apoptosis-Inducing Factor as an efficient nuclease involved in chromatinolysis. PNAS Nexus 2022, 2, pgac312. https://doi.org/10.1093/pnasnexus/pgac312.
6) “In conclusion, this study is to our knowledge the first (…) ApoE and radiation” (lines 684-694). Here, the authors should consider to insert and additional “Conclusions” section and move all these information details.
Reviewer 3 Report
Comments and Suggestions for Authors
The submitted manuscript is of very high quality and I recommend its publication after revisions. I will justify my decision below.
General remarks:
This study aims to investigate the lasting impacts of low-dose GCRsim radiation and gamma rays on the behavior and Alzheimer's brain pathology in elderly female and male mice with APP;E3F, APP;E4F, and WT genetic backgrounds. The aim of the study is valid and important in context of the space travel, which will surely intensify in the upcoming years. The article is also in the scope of IJMS.
The percent match, 20% according to iThenticate report, is a nice result, and is fully accepted. Materials and methods were presented in an excellent way and don’t require any corrections. Number of references, though, is above 100 and thus should be limited solely to the most relevant and recent positions. After all, this is not a review paper.
Introduction is slightly too long, especially part in Lines 67-68 should be shortened
Also, I honestly believe that adding a short Conclusions sections would really increase the clarity of the work and would be beneficial for the article. It can start at line 683.
Minor comments:
There are some minor editorial mistakes and typos, but nothing that can’t be handled by the great MDPI Editorial Team. For example:
References are not formatted properly, i.e. instead of “1. [1] Cucinotta, F.A (…)” it should be “1. Cucinotta, F.A (…)“
Line 1001, here, the authors should list the content of supplementary materials
Lines 70 and 140, it should be “proton”
Line 738, here the authors use “H+” instead of “proton”. While those are synonyms, I would suggest to be consistent.
Concluding:
The research topic, the expressed hypothesis, and the methodology employed in the study exhibit exceptional excellence. It has been quite a while since I last had the chance to read a manuscript without any major remarks or feedback. Congratulations to the authors.
